# Environmental Influences on the Behavioural and Emotional Outcomes of Children: A Network Analysis

**DOI:** 10.3390/ijerph19148479

**Published:** 2022-07-11

**Authors:** Shamshad Karatela, Neil I. Ward, Janis Paterson, Irene Suilan Zeng

**Affiliations:** 1School of Pharmacy, University of Queensland, Brisbane 4006, Australia; 2Department of Chemistry, University of Surrey, Guildford GU2 7XH, UK; n.ward@surrey.ac.uk; 3National Institute for Public Health and Mental Health, Faculty of Health and Environmental Science, Auckland University of Technology, Auckland 0627, New Zealand; janis.paterson@aut.ac.nz; 4Department of Biostatistics and Epidemiology, Faculty of Health and Environmental Science, Auckland University of Technology, Auckland 0627, New Zealand; irene.zeng@aut.ac.nz

**Keywords:** manganese, child behaviour problem, trace elements, children, nail biomarker

## Abstract

Background: Intellectual developmental disorders are a serious source of health morbidity with negative consequences for adults as well as children. However, there is limited evidence on the environmental, trace element, behavioural, and emotional outcomes in children. Here, we investigated whether there is any association between child behaviour and emotional outcomes and micronutrients using network analysis. Methods: A cross-sectional study was conducted in 9-year-old children within a Pacific Island Families study birth cohort. Elemental concentration was determined in children’s toenails after acid digestion and analysed using inductively coupled plasma mass spectrometry. We used network analysis to identify closely associated trace elements and tested the directions and strength of these trace elements. MANCOVA were used to identify the significant associations between individual elements and the behavioural/emotional function of the children using the children behaviour checklist (CBCL). At the final step, quantile regression analysis was used to assess and quantify the identified associations between CBCL function scores and manganese, adjusted by sex, ethnicity, and standardized BMI. Results: Three major nutrient networks were identified. In the Mn network, Mn was strongly positively associated with Al (0.63) and Fe (r = 0.65) and moderately associated with Pb (r = 0.45) and Sb (r = 0.42). Al was also strongly associated with Fe (r = 0.9). Children in the second or third clinical group, with an elevated externalized CBCL score, had a much higher mean and median level of Mn as compared to the normal range group. The aggression score was significantly associated with Mn concentration and sex. Higher Mn concentrations were associated with a higher aggression score. A 1 ug/g unit increase in Mn was associated with a 2.44-fold increase (95% confidence interval: 1.55–4.21) in aggression score, and boys had higher median aggression score than girls (difference: 1.7, 95% CI: 0.9–2.8). Attention and rule breaking scores were both significantly associated with Mn concentration. Higher Mn concentrations were associated with higher attention behaviour problem and rule breaking scores. A 1 ug/g unit increase in Mn was found to be associated with a 1.80-fold increase (95% confidence interval: 1.37–2.82) in attention score, and a 1.46-fold increase (95% confidence interval: 1.01–1.74) in the rule breaking score. Thought score was not significantly associated with Mn concentration (*p* = 0.13) but was significantly lower in boys (*p* = 0.004). Conclusions: Exceeding Mn levels is potentially toxic and has been identified to be associated with worse externalized children’s behavioural health and emotional well-being. Future studies are necessary to find the exposure paths so that advice shall be provided to family and care providers in public health and environmental protection.

## 1. Introduction

Prevalence of intellectual developmental disorders (IDD), such as learning disabilities, attention deficit hyperactivity disorder, developmental delays, and emotional and behavioural problems, are on the rise, affecting 10–15% of all live births globally [1,2]. IDD effects the developing brain, leading to emotional, cognitive, and behavioural processes across the lifespan [3]. Whilst a behavioural problem is multisystemic in nature, certain studies have found that environmental exposures and imbalances in the concentrations of toxic and trace elements are associated with behavioural problems [4,5,6,7]. Trace elements are involved in many biological processes and any alterations can derail the biological system causing lower behavioural, emotional, cognitive, and intellectual function in school children [8,9,10,11,12]. For example, a cohort study found that trace element deficiencies (zinc (Zn) and iron (Fe)) and sociodemographic factors were associated with total behaviour problems in 3- to 5-year-old children [11]. In another cohort study, children as young as 3 years old, with lower Zn and Fe deficiency, protein, and vitamin B, had higher externalizing behaviour problems as compared to the control [13]. Lower dietary intake and low concentrations of docosahexaenoic acid (DHA)-rich oil was associated with aggression in university students [14]. Similarly, a randomized control trial in English adult prisoners, reduced aggression and offending in the group that was given omega 3 multivitamins and mineral supplementation [15]. This is supported by a meta-analysis that concluded that those children with reduced omega 3 had an increased aggression as compared to those with higher omega 3 (SMD = 0.24; ESsg = 0.82; r = −0.09), although it had a small effect size [16]. Children with Fe deficiency, anaemia, and vitamin B-12 showed an association with behaviour problems [17]. A recent study identified that those children who ate higher animal protein had lower externalizing behaviour problems in boys [18]. In a recent review of studies on depression and trace elements, Zn, Fe, and vitamin D supplementation reported decreased depressive symptoms [19].

Environmental exposures such as industrial chemicals and toxic elements (lead (Pb), mercury (Hg), arsenic (As), cadmium (Cd), aluminium (Al), manganese (Mn), and nickel (Ni)), have been linked to IDD [1,20]. Well-known toxic elements like Pb, Cd, and Hg can damage multiple organs even at lower levels in utero and/or postnatally [21,22]. Manganese at lower levels is essential but at higher levels can be toxic, especially in children. Recently, Mn exposure at higher levels has emerged as a developmental neurotoxin. Mn exposures in children mostly occurs via inhalation [23] or ingestion via drinking water [24]. The most hazardous route of Mn exposure is airborne and the particles (≤5 microns) get into the airways and are then absorbed into the blood stream and cross the blood–brain barrier [24,25]. Mn contaminated soil exposure can also occur in children due to their hand-to-mouth exposure [26]. The major source of Mn exposure in adults is occupational (welders mine ore extraction) [27,28] and environmental (mine ore, pesticides); however, this is not the case in children [29].

The general population will usually be exposed to multiple neurotoxins as there may be synergetic, antagonistic, and additive interactions that can occur between these elements [29,30]. However, there is limited evidence on the interactive effects of multiple neurotoxins and trace elements on adverse health outcomes [31]. A recent study found that As and Pb had a synergetic effect to cognitive score and an antagonistic effect between Mn and Hg [29]. A previous cohort study of combined exposure of As and Pb in soil was a risk factor for intellectual disability [32]. A longitudinal study of 455 children in Mexico City reported a synergetic association between Pb and Mn, whereby higher lead toxicity increased with higher Mn exposure [30]. Previously, we studied the impact of heavy metals such as Cr, Pb, Cu, and Ni within Pacific children in relation to smoking [33] and behavioural problems [34]. In another research study, we also demonstrated correlations and interactions between a range of elements within Pacific children [35]. However, in the current study, we aimed to understand the synergetic, antagonistic, and possible additive interactions between toxic elements, trace elements, and other covariates in relation to intellectual development disorders through the network pathway. Furthermore, we explored Mn levels on the behavioural health and emotional well-being of children. The network analysis model showed how the different variables interact and connect providing structural relations among core processes, especially behavioural and emotional well-being in school-aged children.

## 2. Materials and Methods

### 2.1. Study Design and Population

This was a cross-sectional study carried out on the Pacific Island people living in the South of Auckland, New Zealand (NZ). They are the fourth largest ethnic group in NZ at approximately 33.4% of the total population [36].

The study was conducted between July 2010 and July 2011 that recruited a subsample of 9-year-old Pacific children within a birth cohort of the Pacific Island Families (PIF) study who have been followed since birth in the year 2000 [37]. To be eligible within the core PIF study, at least one parent of the newborn child must identify themselves as being of one of the Pacific Island ethnicities and as a permanent resident of NZ. The children (age 9 years) for this sub-study who were healthy were enrolled at the same time as the main PIF cohort data collection wave.

Consent and assent were sought from children and their mothers before commencing this study. Children who provided consent but had short nails (typically less than 0.05 g of cut material) were excluded. This study was approved by the NZ Health and Disability Ethics Committee (NTX/07/05/050).

### 2.2. Toenail Sample Collection and Laboratory Analysis

All toenail samples were washed prior to digestion to remove potential exogenous elemental contaminants derived from cosmetic treatments, ‘dirt’, etc. The washing procedure involved: (1) five steps using acetone, deionized distilled water (DDW, 18.2 MΩ) (×3), and then acetone again; (2) at each washing step enough liquid was added to cover the sample and sonication (ranssonic water bath (T460/H)) for 5–10 min; and (3) decantation. Following the washing procedure, the nail samples were dried overnight at 60 °C in a drying oven (LTE Scientific, Oldham, UK). Once dried, the sample was weighed (four decimal place analytical balance) and transferred to a pre-acid/DDW washed/dried Kjeldhal ™ tube for digestion, where 0.5 mL of concentrated nitric acid (Fisher Scientific, Waltham, MA, USA, Trace Analysis Grade nitric acid) was added, and the tube was sealed with PVC Clingfilm. The Kjeldhal ™ tube was placed in a hot block (Tecator 2012 Digestor) and heated at 160 °C for 30–60 min. Once the digestate was visibly clear, the Kjeldhal ™ tube was removed from the heat and cooled and the digest solution transferred to a clean, weighted 15 mL centrifuge tube. The digested sample was weighed again (to four decimal places) and diluted 250 times (volume/weight) using DDW based on a dilution factor of 250. Due to some small toenail sample masses the 250-dilution factor was not sufficient to ensure there was enough volume of sample (4 mL) to be analysed; thus, for some samples, a higher dilution factor was used and noted. Before analysis, the digest was filtered using a 0.22 μm syringe-driven filter unit (Millex^®^-GP, Millipore, Bedford, MA, USA). Analysis of all the washed and digested nail samples was carried out by the Agilent 7700 × ICP-MS instrument (Agilent Technology, Didcot, UK). Two certified reference materials (CRM) were used, namely, NIST SRM 1643e (National Institute of Standards and Technology, Gaithersburg, MD, USA) and TMDA-54.4 (National Water Research Institute, Canada). The recovery rate range for all the elements was between 76% and 101.3%. All instrumental data for each element (according to the isotope selected) were reported as counts per second. The value was corrected for a reagent blank signal (to correct for any contribution from the digestion procedure) and ratioed with the internal standard isotope value (to correct any instrumental drift or signal enhancement/depression caused by the matrix). Data for the calibration standards were handled in the same manner and an Excel™ calibration curve produced for each element, with a ratio signal (*y*-axis) and concentration of five standards (*x*-axis), from which the calibration equation was determined for calculation of the unknown toenail sample elemental concentration. The elemental values for each toenail sample were corrected for the dilution factor and the final values used in data analysis.

### 2.3. Questionnaires

Validated and reliable questionnaires were administered at the 9-year phase to both mother and child. These questionnaires were interviewer administered regarding socio-demographic, cultural, environmental, child development, family, and household dynamics, as well as lifestyle and health issues. Participant characteristics and demographic variables that were of interest for this study were included in the analysis (child’s sex (girls, boys) and child Pacific ethnicities (Samoans, Tongan, Cook Island, and Others (which included Tokelau and Niuean), as determined by the mother at the age of 2 years).

The child behaviour checklist (CBCL) [38], now called the Achenbach System of Empirically Based Assessment (ASEBA), was used to detect behavioural and emotional problems of children [39]. This was parent reported to screen for emotional, behavioural, and social problems. The parental version of the 120-item Child Behaviour Check List (CBCL)/6-18 was utilized for the maternal PIF study participants regarding the behaviour of their 9-year-old children and was administered by the PIF interviewers. The CBCL questionnaire was completed by the mothers or the child-carers. The CBCL is assessed on a 3-point Likert-type scale: 0 = not true, 1 = somewhat or sometimes true, and 2 = very true or often true. The time frame for item responses was within the past six months. The CBCL included overall total problem scores (T-scores); two broad-band syndrome scores, internalising and externalising; and seven narrow-band syndromes: emotionally reactive, anxious/depressed, withdrawn, somatic complaints, sleep problems, attention problems and aggressive behaviour [38]. The CBCL is classified as being in the clinical, borderline, and normal range with higher scores indicating greater degrees of behavioural and emotional problems.

### 2.4. Anthropometric Measurement

Obesity was measured by standardized Body Mass Index (BMI), height-to-waist ratio, and percentage body fat. Prior to data collection, the equipment was standardized, procedures were documented in an operation manual, and assessors were trained. The average weight and height were calculated. Average weight and height were used to calculate children’s BMI as weight (kg) divided by height squared (m^2^).

### 2.5. Statistical Analysis

All data, which were obtained for this sub-study, were stored in Microsoft AccessTM and ExcelTM (Microsoft Corporation, Redmond, WA, US). The data were edited, range and consistency checks were performed, and the data were coded for analysis when necessary. Questions with no response received a distinct code and were not included in the analysis. The edited data were then exported to a statistical software, SAS and R.

Network analyses (R package “huge” and “glmnet”) were used to identify possible solutions of trace element networks. The solution is selected by a panelized parameter lambda that is used to control the density of the network using the glasso approach. The solution with mid-point value of lambda was selected to balance between simplicity and interpretability. The identified networks were verified and checked using the intra correlation coefficient matrix of each network. Directions of the linkage were determined by the sign (+/−) of the Spearman correlation coefficients. The R package “corrplot” was used to create the correlation heatmap.

A generalized linear model (GLM) was used to assess the association between ordinal externalized CBCL outcome (normal/borderline/clinical) and Mn concentration of the toenail sample.

Before the confirmative analysis, MANOVA were used within each identified element network to evaluate significant associations between element and CBCL score (total score, externalized factor, and internalized factor). Multivariate linear regressions were used to analyse the associations between multiple CBCL externalized syndrome subscales (natural log transformed due to skewness) and trace elements (Mn, Fe, Cu), adjusted by gender, ethnicity, and standard BMI. Multiple linear regression was used to evaluate the total CBCL externalize score and Mn, adjusted by the same set of demographic and growth variables.

In confirmative analysis, quantile (median) regressions were applied to assess the association of CBCL externalized factors (aggression, rule breaking, and attention) and Mn adjusted by gender, ethnicity, and standard BMI [34,35]. Generalized logistic regression was applied to thoughts and Mn with the same confounding variables. Element intensities were natural log transformed in all the above regression analysis.

SAS^®^ proprietary software 9.4 and R version 4.0.2 were used in the analysis.

## 3. Results

### 3.1. Demographics

In total, 278 9-year-old Pacific Island children were included in the study. There were 118 (42%) girls and 160 (58%) boys, and most of them were of Samoan ethnicity (148: 53%); the other ethnicities were Tonga (29: 10%), Cook Island (53: 19%), and Other (48: 17%). The average age of mothers at birth of these children was 28 years.

The proportion of children with behavioural problems in the clinical range (61/278 (22%): 95% CI: 17% to 27%) was higher than expected given that the “clinical range” was defined as having a CBCL total problem score above the population 90th percentile. The mean CBCL of the study cohort is 27.2 (std: 13.5), and the median is 25 with interquartile range 17–36.

### 3.2. Descriptive Summary

As shown in Table 1, the fourth externalizing CBCL quartile group of children participants had a median Mn level of 4.0 µg/g dry weight (25–75% quartile, 2.3–8.2) and a mean of 7.2 ug/g (std: 10.2) d.w. Using the clinical defined threshold (Table 2), children participants were either in the typically developed group (scored in the normal range and below the clinical range) or the clinical group (scored within the clinical range). Children that fell within the clinical range were classified as having child behaviour problems. Children in the second or third clinical group had a much higher mean and median level of Mn as compared to the typically developed group (median: 2.9, 6.4, and 7.2 µg/g d.w. of the normal range, the second, and third clinical groups, respectively (Figure 1); the linear trend test of mean Mn on a log scale results in a *p* value of 0.001).

### 3.3. Network Analysis

Using network analysis, the three major correlation networks’ solution among the trace elements was identified (Figure 2). In the first network of 10 elements, Mn had the most significant linkages with the other 7 elements (Al, Fe, Sb, Cu, B, As, and Pb). In the second (Se, Mo, Hg, and I) and third (Zn, Ca, Cd, and Mg) networks, each element was evenly connected to each other. Co and Ni were linked as the fourth network.

In the Mn network, Mn was strongly positively associated with Al (Pearson correlation coefficient r: 0.63) and Fe (r: 0.65) and moderately associated with Pb (r: 0.45) and Sb (r: 0.42). Al was also strongly associated with Fe (r: 0.9). All correlations are statistically significant at the 0.05 level (Figure 3).

In the Zn network, Cd was found to be strongly negatively associated with Mg, Zn, and Ca. Mg, Zn, and Ca were strongly positively associated with each other (Table 3).

In the Se network, Se was strongly positively associated with Mo, but negatively associated with Hg and I. Hg and I were strongly positively associated with each other (Table 4).

### 3.4. Associations between CBCL, Mn Concentration and Other Factors

Both the ANCOVA and multivariate analysis of variance (MANOVA) indicate that the Mn concentrations were significantly associated with the external CBCL score and its syndrome subscales. The external CBCL score was also significantly different between boys and girls. Quantile (median) regression of each CBCL external syndrome subscale confirmed similar findings, which are summarized as follows.

The aggression score was significantly associated with the Mn concentration and gender. A higher Mn concentration was associated with a higher aggression score; boys had a higher median aggression score than girls (difference: 1.7, 95% CI: 0.9–2.8). A 1 µg/g unit increase in Mn level was associated with a 2.44-fold increase (95% confidence interval: 1.55–4.21) in the median aggression score.

The attention and rule breaking scores were both significantly associated with the Mn concentration. A higher Mn concentration was associated with a higher attention and rule breaking score. A 1 µg/g unit increase in Mn was associated with a 1.80-fold increase (95% confidence interval: 1.37–2.82) in the median attention score and a 1.46-fold increase (95% confidence interval: 1.01–1.74) in the median rule breaking score.

The thought score was not significantly associated with the Mn concentration (odds ratio: 0.86, 95% confidence interval 0.70–1.05, *p* = 0.13) but was significantly lower in boys than girls. The boys’ thought score was 44% (95% confidence interval: 27–69%) less likely and at a higher score than the girls.

## 4. Discussion

This was a first study that has used the network analysis to understand the effects of the environment, trace elements, and neurodevelopment in 9-year-old Pacific Island children living in NZ. Network analysis is a statistical approach that investigates relationships between variables. This analysis allowed us to explore various variables and reveal the complex patterns of relationships and network structures [40]. Despite our knowledge on how elements work to support our physiology, there is a limited knowledge on how these elements interact with each other and what this means in terms of health. In our study, there were three major networks and one minor network that were identified between the elements under investigation.

### 4.1. Network Analysis

In the first network, manganese (Mn) had the most significant linkages with toxic elements (Al, Sb, As, and Pb) and the trace elements (Fe, Cu, and B). Additionally, within our study, Mn was positively associated with Al, Fe, Pb, and Sb, which means an increase in Mg levels is associated with increases in Al, Pb, Sb, and Fe. All of these elements at higher levels can be toxic, and some like Pb can bioaccumulate in humans. Manganese is an essential nutrient. It has a role in the growth and development of humans, especially in metabolic reactions such as the glycolytic cycle, the citric acid cycle, and the beta oxidation of fatty acids [41]. However, there is growing evidence that high environmental concentrations of Mn can be toxic [42] and can accumulate in the brain [43]. Iron (Fe) is an essential trace element that is needed for our immune system and for energy production [44]; however, excessive blood Fe levels can cause cirrhosis and fibrosis as well as cardiac problems [45]. Sufficient levels of Cu are required in the transportation and mobilisation of iron stores, and hence, a deficiency of Cu would cause hepatic iron overload [45]. There is evidence building on the fact that B may be essential for humans [46]. It has been shown to be beneficial in bone growth and maintenance, improved central nervous system, inflammation and oxidative stress, and cancer reduction [47]. However, the toxic effects of B can occur at cellular levels [48]. Furthermore, the network identified that, for Fe, Mn, and Al in our study, these elements are known to be found in sediments and acidic soils [49], in air, water, and living things as well [50]. Together they play a role in the absorption and transport of certain essential and toxic trace elements [50]. However, the pathways and occurrence of this network in humans is an area that requires further exploration as not very much is known about this network in people, especially children. Excessively high Mn levels and Pb have a similar mechanism in that they both are involved in the disturbance of calcium metabolism and release of neurotransmitters, affecting the central nervous system, especially in growing children [30]. A study by Henn et al. in their study observed a synergetic effect between Pb and Mn, whereby increased Mn levels increased the Pb concentration [30]. This is similar to our current research results for the Mn network. Arsenic also shares the same neurotoxic effects as Pb and high Mn concentrations, and hence, would have a similar adverse neurological effect on children, which means these elements all have synergetic effects [51]. The positive interaction between Mn, Pb, and As was found within the Mn network. The data reveal that Mn plays roles in both the known toxic element subnetwork (Al, Sb, As, and Pb) and trace elements subnetwork (Fe, Cu, and B). We are not certain if the latter subnetwork elements also add to possible toxic effects, because there were only marginal statistically significant associations between the concentrations of these elements and the external CBCL in our study results (p values range between 0.06 to 0.08 in the ANOVA analysis).

In the Zn network, higher Cd levels are associated with lower Mg, Zn, and Ca levels. Research has shown that the toxic element Cd interacts with many essential elements like Zn, Ca, Se, and Fe [52,53,54]. Similar to our results, Cd seems to have a correlation with Ca as reported by Brzóska and Moniuszko-Jakoniuk [54]. Additionally a review reported that Cd had an interaction with Zn, whereby Zn deficiency can increase Cd accumulation, leading to toxicity [52]. It has also been shown that Mg has a protective effect against Cd [55]. In our Zn network it was observed that Zn and Cd and Mg and Cd have a pairwise antagonistic association, and therefore, Zn and Mg supplementation may be useful addressing the higher levels of Cd and lower levels of specific essential elements.

In the Se network, a positive association was observed with molybdenum but was negatively associated with I and Hg. It is known that Se and Mo (along with other essential elements) are involved in many biological processes in the human body [56]. The toxic element Hg has been known to neurotoxin like Pb. The interaction between Se and Hg is well known, whereby Se has a protective effect against methylmercury [57]. Our Se network showed that higher levels of Hg were associated with lower levels of Se, which is similar to other studies [58].

### 4.2. Manganese Concentrations

Manganese is an essential nutrient and needed for the growth and development in humans; however, overexposure can cause neurobiochemical disturbances leading to nervous or neurodevelopmental disorders, particularly in growing children [59,60]. The mean Mn levels in our study were higher than the reported for the mean Mn in a children study (aged 6–9 years) in an Italian study. They reported the absolute concentrations of metals (µg/g) in toenails from children living within a 3 km radius circle around two incinerators (exposed area). In their study, the reported absolute concentration of Mn was 4.40 ± 1.23 µg/g for children in exposed area and 2.47 ± 0.3 µg/g for children living in an urban area [61]. The third externalizing CBCL quartile group had a similar mean level of 4.8 µg/g Mn compared to this Italian study. Furthermore, a differential pattern was observed in our study between boys and girls. This is similar to other reports, such as a study conducted in California where boys had better cognitive function than the girls [62]. Moreover, opposite effects in another study stated that boys had poorer cognition when compared to girls [63]. Animal studies have shown that males and females accumulate Mn in different body tissues [64]. The differences in gender observed in this study could be due to the kinetics and metabolism that effects girls differently to boys, and therefore, different cut-offs of CBCL level are necessary for different genders [65].

### 4.3. Manganese Concentrations and Behaviour Problems

Our study shows that the children in the clinical range of behaviour problems had higher Mn levels than those in the normal behaviour group. Studies have reported higher Mn levels in relation to hyperactivity [66] and externalising child behaviour problems [67,68]. Studies have reported that excessive exposure to Mn over a two-year period resulted in cognitive, motor, and neurological disorders [68,69,70,71]. Additionally, children’s motor, cognitive, and behavioural functions are affected by Mn exposure [42,72] similar to the current study. In a Canadian study, there was a strong correlation between hair Mn and hyperactivity in children exposed via drinking water [65]. Furthermore, exposure during pregnancy has shown to have an adverse effect on the growing fetus [72]; however, the prenatal effect in our study is not known. A cross-sectional study conducted in Bangladesh on school children reported a reduction in the mathematics achievement scores in those children that drank Mn-exposed water [73]. Airborne Mn exposure was inversely associated with diminished intellectual function in 7- and 11-year-old children living in the Molango mining district in Mexico [28]. Effects on motor functions, attention, and memory have also shown to be affected due to higher Mn levels (ingested via drinking water), in 6- to 13-year-old Canadian children [74]. Other studies in areas where children were exposed to a ferro-manganese plant had high Mn levels and found an association with externalising behaviour [75], similar to our study. In a Brazilian study, children who had higher hair Mn levels in the exposed group showed an association with hyperactivity than the non-exposed group [76]. Another similar study observed that toenail Mn levels showed an association with total behaviour problems in children [23]. Adverse effects on externalizing, internalizing, and hyperactivity due to higher dentine Mn exposure pre- and postnatally was observed in 7- and 10-year-old children [62]. Interestingly, in this study, they also observed a positive association only in the boys cognitive and motor skills that were exposed to higher levels of Mn pre- and postnatally [62] Further research is required to understand Mn exposure at higher levels and its effect on children’s positive or negative behaviour.

### 4.4. Strengths and Limitations

The strength of this study is that we used toenails as a biomarker for manganese and all the other elements measured, as toenails reflect a cumulative exposure over the 7 to 12 previous months [77] and the results are reproducible [78]. Additionally, toenails measure a long-term exposure since toenails grow much slower than fingernails, accumulating the elements over longer time period [78]. Since our study was conducted in children, nails were most appropriate as they are a non-invasive tissue. A limitation of this study is the cross-sectional study design, which does not provide causality. Additionally, all the children were of one age, so we do not know whether behaviour changes are worse in younger or older children. Within our study, we do not know how the children are exposed to Mn, which could be via diet, air, or water. Further investigations are required to explore these topics.

## 5. Conclusions

Our results suggest that high Mn levels in child toenails are associated with externalising and aggressive behaviour. It is important to understand how elements interact with each other and what their effects are in combinations rather than as single elements and the various pathways. This research identified toxic and essential element networks of Mn. More research is required in understanding different networks of elements in humans (for different tissues or fluids as biomarkers) and their impact so that a comprehensive and holistic approach can be taken to guide in addressing the nutritional intake and health care of children.

## Figures and Tables

**Figure 1 ijerph-19-08479-f001:**
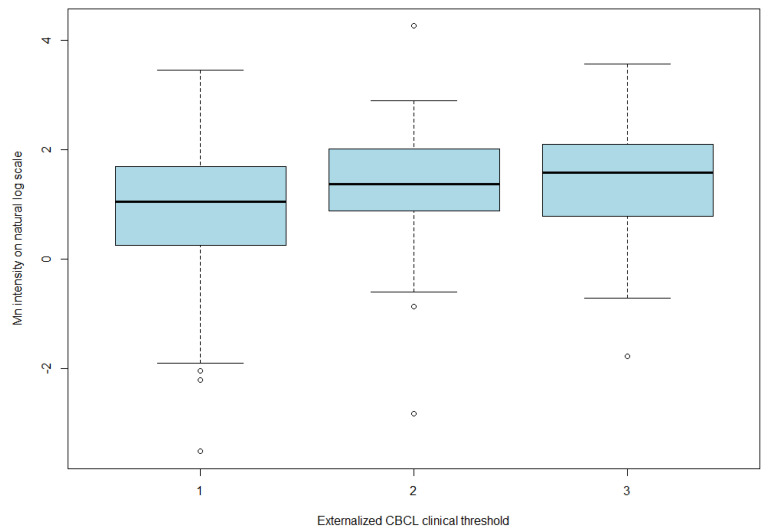
Mn intensity on natural log scale by clinical group of externalized CBCL.

**Figure 2 ijerph-19-08479-f002:**
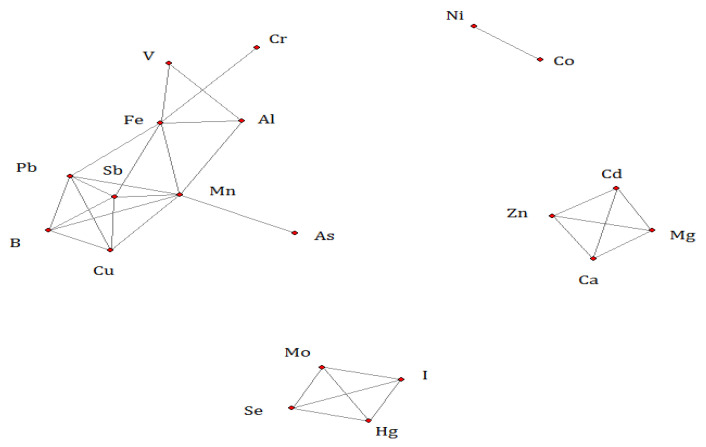
A Gaussian network solution of trace elements (on the natural log scale of the concentration level).

**Figure 3 ijerph-19-08479-f003:**
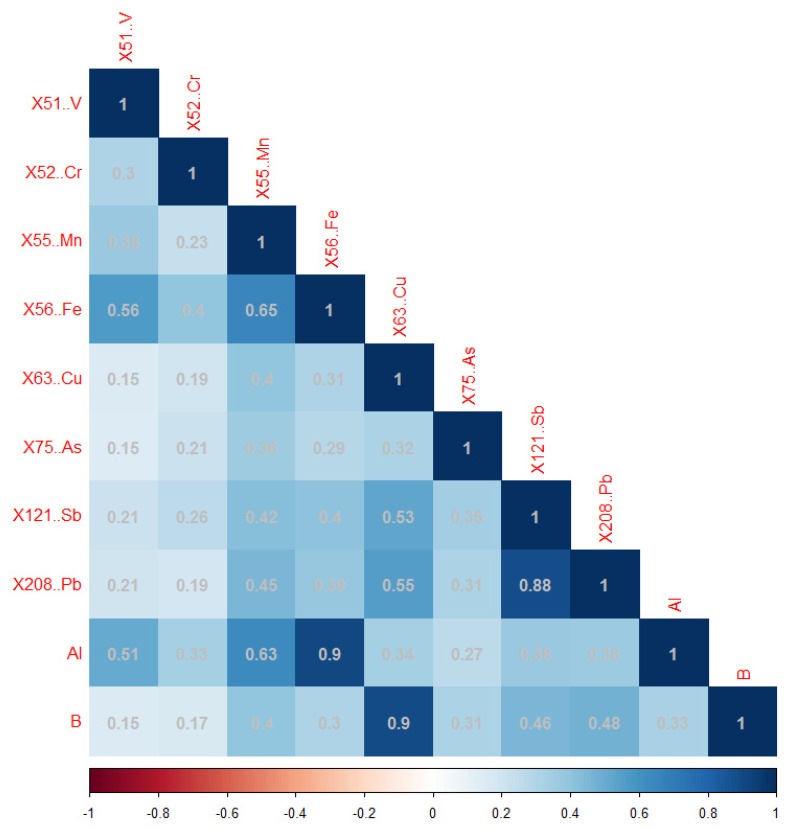
The correlation heatmap of trace element concentrations in the Mn network (on natural log scale).

**Table 1 ijerph-19-08479-t001:** Distribution of Mn by quartile group of externalizing CBCL scores of Pacific Island Families (PIF) 9-year-old children (µg/g, dry weight) and by clinical defined group (normal/within clinical range).

**(A)**									
**Externalized CBCL Quartile**	**Children (n)**	**Mean**	**Std Dev**	**Geo-mean**	**Min**	**Max**	**Median**	**25th Percentile**	**75th Percentile**
1	69	3.9	4.4	2.5	0.1	31.3	2.8	1.4	5.4
2	85	3.7	3.7	2.2	0.0	16.7	2.6	1.1	4.7
3	51	4.8	4.0	3.3	0.4	18.1	3.5	1.5	6.9
4	68	7.2	10.2	4.0	0.1	70.3	4.0	2.3	8.2
**(B)**									
**Externalising Clinical Range**	**Children (n)**	**Mean**	**Std Dev**	**Geo-mean**	**Min**	**Max**	**Median**	**25th Percentile**	**75th Percentile**
1	190	3.9	3.9	2.5	0.03	31.3	2.9	1.3	5.4
2	47	6.4	10.3	3.6	0.1	70.3	3.9	2.3	8.0
3	40	7.2	7.9	4.3	0.2	35.1	4.9	2.2	8.1

(**A**) Trend test *p* value is 0.0025; min—minimum, max—maximum. (**B**) Trend test *p* value is 0.001, Geo-mean—Geometric mean, min—minimum, max—maximum.

**Table 2 ijerph-19-08479-t002:** Distribution of Fe, Cu, and boron by quartile and clinical defined group of externalizing CBCL scores of PIF 9-year-old children (µg/g, dry weight).

	**Quartile Groups:**								
	**Fe:**								
**Externalized CBCL Quartile**	**Children** **(n)**	**Mean**	**Std Dev**	**Geo-mean**	**Min**	**Max**	**Median**	**25th** **Percentile**	**75th** **Percentile**
1	69	81.4	57.2	65.3	13.7	246.4	62.9	39.1	106.2
2	85	114.8	215.8	70.5	16.0	1868.7	60.2	40.4	116.3
3	51	117.2	84.9	93.5	22.7	403.0	95.5	55.4	148.7
4	68	107.0	115.9	79.0	17.5	823.8	76.3	50.9	114.3
	**Cu:**								
**Externalized CBCL Quartile**	**Children** **(n)**	**Mean**	**Std Dev**	**Geo-mean**	**Mini**	**Maxi**	**Median**	**25th** **Percentile**	**75th** **Percentile**
1	69	18.6	9.0	17.04	5.4	64.6	17.0	13.0	21.1
2	85	18.0	9.9	16.47	7.8	78.0	15.5	13.0	20.5
3	51	18.4	6.7	17.36	9.1	41.3	17.3	13.8	21.6
4	68	22.0	14.5	19.34	2.5	116.8	19.0	14.4	25.6
	**B:**								
**Externalized CBCL Quartile**	**Children** **(n)**	**Mean**	**Std Dev**	**Geo-mean**	**Mini**	**Maxi**	**Median**	**25th** **Percentile**	**75th** **Percentile**
1	69	0.29	0.13	0.27	0.12	0.87	0.27	0.22	0.32
2	85	0.28	0.13	0.27	0.12	0.99	0.27	0.21	0.32
3	51	0.28	0.11	0.27	0.16	0.67	0.28	0.21	0.32
4	68	0.33	0.14	0.30	0.09	0.78	0.28	0.24	0.38
	**On Clinical Defined Groups:**								
	**Fe:**								
**Externalising** **Using CBCL Criteria**	**Children** **(n)**	**Mean**	**Std Dev**	**Geo-mean**	**Mini**	**Maxi**	**Median**	**25th** **Percentile**	**75th** **Percentile**
Normal	190	103.6	153.4	72.5	13.7	1868.7	65.4	42.7	121.3
Borderline	47	102.8	76.8	83.2	23.0	369.3	83.5	54.1	117.7
Clinical	40	114.7	138.8	79.9	17.5	823.8	74.7	48.7	121.9
	**Cu:**								
**Externalising** **CBCL Criteria Range**	**Children** **(n)**	**Mean**	**Std Dev**	**Geo-mean**	**Mini**	**Maxi**	**Median**	**25th** **Percentile**	**75th** **Percentile**
Normal	190	18.2	8.9	16.8	5.4	78.0	16.3	13.1	21.1
Border line	47	21.3	9.3	19.7	11.9	54.4	18.9	14.5	26.2
Clinical	40	22.0	17.2	18.7	2.5	116.8	18.6	13.9	24.7
	**B:**								
**Externalising** **CBCL Criteria**	**Children** **(n)**	**Mean**	**Std Dev**	**Geo-mean**	**Mini**	**Maxi**	**Median**	**25th** **Percentile**	**75th** **Percentile**
Normal	190	0.29	0.12	0.27	0.12	0.99	0.27	0.21	0.32
Border line	47	0.32	0.14	0.30	0.17	0.76	0.28	0.22	0.38
Clinical	40	0.32	0.14	0.30	0.09	0.78	0.29	0.24	0.38

**Table 3 ijerph-19-08479-t003:** The Pearson correlation matrix of the Zn network elements (on natural log scale).

	Cd	Mg	Zn	Ca
Cd	1			
Mg	−0.71	1		
Zn	−0.76	0.86	1	
Ca	−0.74	0.89	0.97	1

**Table 4 ijerph-19-08479-t004:** The Pearson correlation of Se network elements (on natural log scale).

	Se	Mo	Hg	I
Se	1			
Mo	0.91	1		
Hg	−0.94	−0.89	1	
I	−0.76	−0.73	0.78	1

## Data Availability

Not applicable.

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
