# Peer review of "Environmental Influences on the Behavioural and Emotional Outcomes of Children: A Network Analysis"

_ijerph, 2022, doi:10.3390/ijerph19148479_

Round 1
Reviewer 1 Report
This is interesting research on behavioral effects from environmental exposure to manganese among children. Authors also analyze the network correlation of trace elements. The manuscript is overall well written with sufficient level of details but need some improvement specially at the discussion section.
Line 32 - with higher attention – correct to attention behavior problem
Introduction
Line 75 - Burton, N.C.; T.R. Guilarte. Manganese Neurotoxicity: Lessons Learned from Longitudinal Studies in Nonhuman Primates. 458 Environmental Health Perspectives 2009, 117, 325–332 - This reference is not related to children and Mn exposure - need to be removed. You can add the following Liu, W., Xin, Y., Li, Q., Shang, Y., Ping, Z., Min, J., Cahill, C. M., Rogers, J. T. & Wang, F. (2020). Biomarkers of environmental manganese exposure and associations with childhood neurodevelopment: A systematic review and meta-analysis. Environmental Health: A Global Access Science Source, 19(1), 1–22. https://doi.org/10.1186/s12940-020-00659-x
Line 93 – correct Pd to Pb
Line 95 - to be consistent use always Pb (lead)
Methods
CBCL – Provide an explanation related to the classifications that CBCL has (clinical, borderline and non-clinical), because it is used in the results section without any explanation previously.
Line 201 - total score, externalized factor and internalize factor – the names of factors are internalizing and externalizing , need to be consistent in all the manuscript.
Line 203 – “adjusted by gender, ethnicity, standard BMI” – If the set of demographic variable and growth variable were selected based on previous literature it should be mentioned or provide the explanation of this selection.
Results
Table 1 and Table 2:
External_9y_cat - what mean this? Seams to be the variable name, could be change only to the name of the variable
using clinical threshold (1,2 and 3) - Its not clear this 3 groups. I suppose the authors are using the 3 categories of CBCL (non-clinical; borderline and Clinical). This need to be included at methods section and at the final text of the table (comments below table).
Table 3. Pearson correlation matrix of the Mn network elements. Please provide indications if all the correlations were significant. Why only 3 correlations are in bold? The authors need to provide information. Same comment to the following tables that provide data about correlations.
Regarding CBCL MANOVA results, it would be better to have at least one table summarizing the results stead to present it only describing on the text.
Discussion
Line 315 - neurotic effects - I think authors mean neurotoxic effects
Line 342 - animals However – need point
The discussion related to manganese concentrations and child cognition and behavior need to be improved. Specially need to discuss more the finds related to externalizing scores and manganese concentrations and differences in gender since the literature in this topic are large. The discussion section needs to be broader and discussed in terms of its specific results, such as the association between Mn levels and externalizing behavior in boys. For example, some studies found more impact in girls, some studies didn’t.
Some studies already mentioned could be addressed broadly like Rodrigues et al 2018, Mora et al 2015, Menezes et al 2014, Bouchard 2007. Also other studies could be included to discuss the association between manganese exposure, externalizing behavior and attention behavior problems.
Carvalho, C. F., Oulhote, Y., Martorelli, M., Carvalho, C. O. de, Menezes-Filho, J. A., Argollo, N. & Abreu, N. (2018). Environmental manganese exposure and associations with memory, executive functions, and hyperactivity in Brazilian children.
Oulhote, Y., Mergler, D., Barbeau, B., Bellinger, D. C., Bouffard, T., Brodeur, M.-È., Saint-Amour, D., Legrand, M., Sauvé, S. & Bouchard, M. (2014). Neurobehavioral function in school-age children exposed to manganese in drinking water.
Author Response
Reviewer: Line 32 - with higher attention – correct to attention behavior problem
Response: Thanks for the corrections. We have now fixed it.
Introduction
Reviewer: Line 75 - Burton, N.C.; T.R. Guilarte. Manganese Neurotoxicity: Lessons Learned from Longitudinal Studies in Nonhuman Primates. 458 Environmental Health Perspectives 2009, 117, 325–332 - This reference is not related to children and Mn exposure - need to be removed. You can add the following Liu, W., Xin, Y., Li, Q., Shang, Y., Ping, Z., Min, J., Cahill, C. M., Rogers, J. T. & Wang, F. (2020). Biomarkers of environmental manganese exposure and associations with childhood neurodevelopment: A systematic review and meta-analysis. Environmental Health: A Global Access Science Source, 19(1), 1–22. https://doi.org/10.1186/s12940-020-00659-x
Response: We thank the reviewer for the suggestions.
We have now added the reference “Liu, W., Xin, Y., Li, Q., Shang, Y., Ping, Z., Min, J., Cahill, C. M., Rogers, J. T. & Wang, F. (2020). Biomarkers of environmental manganese exposure and associations with childhood neurodevelopment: A systematic review and meta-analysis. Environmental Health: A Global Access Science Source, 19(1), 1–22. https://doi.org/10.1186/s12940-020-00659-x” . It is still reference 27 but with the above.
We have taken out the “Burton Burton, N.C.; T.R. Guilarte. Manganese Neurotoxicity: Lessons Learned from Longitudinal Studies in Nonhuman Primates. 458 Environmental Health Perspectives 2009, 117, 325–332” reference
Reviewer: Line 93 – correct Pd to Pb
Response: Thanks for the correction. We have fixed it to Pb.
Reviewer: Line 95 - to be consistent use always Pb (lead)
Response: We thank the reviewer. We have made sure its all Pb to be consistent
Methods
Reviewer: CBCL – Provide an explanation related to the classifications that CBCL has (clinical, borderline and non-clinical), because it is used in the results section without any explanation previously.
Response:
We thank the reviewer for this important correction. We have now added this:
“The CBCL is classified into the clinical, borderline and normal range with higher scores indicating greater degrees of behavioral and emotional problems”.
Reviewer: Line 201 - total score, externalized factor and internalize factor – the names of factors are internalizing and externalizing , need to be consistent in all the manuscript.
Response: We have now corrected this as noted by the reviewer.
Question: Line 203 – “adjusted by gender, ethnicity, standard BMI” – If the set of demographic variable and growth variable were selected based on previous literature it should be mentioned or provide the explanation of this selection.
Response: The selected variables are based on previous study [42,43]
Results
Reviewer: Table 1 and Table 2:
External_9y_cat - what mean this? Seams to be the variable name, could be change only to the name of the variable
Response: We have now fixed so that it reads as “Externalising CBCL criteria”
Reviewer: using clinical threshold (1,2 and 3) - Its not clear this 3 groups. I suppose the authors are using the 3 categories of CBCL (non-clinical; borderline and Clinical). This need to be included at methods section and at the final text of the table (comments below table).
Response: The reviewer is right. This should be Normal, borderline and clinical range. We have fixed it.
Reviewer: Table 3. Pearson correlation matrix of the Mn network elements. Please provide indications if all the correlations were significant. Why only 3 correlations are in bold? The authors need to provide information. Same comment to the following tables that provide data about correlations.
Response: All the correlation are statistically significant. This information is added in the result.
Reviewer: Regarding CBCL MANOVA results, it would be better to have at least one table summarizing the results stead to present it only describing on the text.
Response: The MANOVA and other regressions are all parts of the analysis before the final confirmative data analysis. The final confirmative analysis used the quantile regression and results are interpreted in the text.
Discussion
Reviewer: Line 315 - neurotic effects - I think authors mean neurotoxic effects
Response: We thank the reviewer for the correction. We have now fixed it to neurotoxic effects.
Reviewer: Line 342 - animals However – need point
Response: We thank the reviewer. The sentence has been corrected and it now reads as follows:
“Manganese is an essential nutrient and needed for the growth and development of animalsin humans, however, over exposure can cause neurobiochemical disturbances leading to nervous or neurodevelopmental disorders, particularly in growing children [66, 67]”
Reviewer: The discussion related to manganese concentrations and child cognition and behavior need to be improved. Specially need to discuss more the finds related to externalizing scores and manganese concentrations and differences in gender since the literature in this topic are large. The discussion section needs to be broader and discussed in terms of its specific results, such as the association between Mn levels and externalizing behavior in boys. For example, some studies found more impact in girls, some studies didn’t.
Some studies already mentioned could be addressed broadly like Rodrigues et al 2018, Mora et al 2015, Menezes et al 2014, Bouchard 2007. Also other studies could be included to discuss the association between manganese exposure, externalizing behavior and attention behavior problems.
Carvalho, C. F., Oulhote, Y., Martorelli, M., Carvalho, C. O. de, Menezes-Filho, J. A., Argollo, N. & Abreu, N. (2018). Environmental manganese exposure and associations with memory, executive functions, and hyperactivity in Brazilian children.
Oulhote, Y., Mergler, D., Barbeau, B., Bellinger, D. C., Bouffard, T., Brodeur, M.-È., Saint-Amour, D., Legrand, M., Sauvé, S. & Bouchard, M. (2014). Neurobehavioral function in school-age children exposed to manganese in drinking water.
Response: We thank the reviewer for their suggestions. We have now added more to the discussions section and used the references kindly suggested by reviewer.
Reviewer 2 Report
This analysis is timely considering the lack of previous data on biomarkers of exposure in this particular demographic. However, there are several questions about how the analysis was conducted and how the data are presented. Additionally, there are some grammatical errors that could be easily addressed with a more thorough proofread. Please see my detailed comments as follows:
Abstract:
In the sentence stating “Children in the second or third clinical group, with an elevated externalized CBCL score, had a much higher mean and median level of Mn as compared to the normal group. The aggression score was significantly associated with Mn concentration and gender,” please provide statistics. If word limits are a barrier, I see several sentences that should probably be combined.
Please also provide statistics for the statement “Thought score is not significantly associated with Mn concentration but is significantly lower in boys.”
The abstract states that this analysis was adjusted for ethnicity, but the methods section indicates that there was only one ethnic group enrolled in the study.
The authors toggle back and forth between presenting their findings in present tense and past tense. Please present all results in past tense.
The abstract also states that the analysis was adjusted for gender, but the methods indicate that it was adjusted for sex. Gender and sex are not the same and should not be used interchangeably. Are the authors referring to sex assigned at birth? If so, sex should be used.
Introduction:
Please revise the following sentence to either focus on symptomology or proposed etiology. As it stands, it is too confusing what the sentences is supposed to convey: “interrelation between the brain, emotional, cognitive, genetics and behavioral processes across the lifespan”.
Manganese is also an essential element, so it can be hazardous when exposures are too low. This should be conveyed as it is only toxic at high doses.
The authors suggest that the major source of manganese is occupational exposure, but that’s only in adults. As this study focuses on 9 year old children, that is not a relevant exposure source. The primary source of manganese in children in nutritional intake, particularly infant formula.
The introduction currently provides far too much study specific information that could simply be moved to the discussion. Please use the intro to briefly summarize what is known and why this study was conducted.
Methods:
It is not clear why the authors used MANOVA when the score is ordinal and not continuous.
The utility of network analysis should be described for understanding the reasons for use over other methods of join associations (WQS, g-computation, PCA).
Additionally, just because the data are coded in an ordinal fashion does not make them ordinal. Have the authors tested the proportional odds assumption to confirm that they data are actually ordinal? If not, they should be modeled as nominally discrete variables.
If the authors are including multiple predictors in the model, that is a “multivariable” model, not “multivariate”.
The authors state the data were transformed, but there are several ways to transform the data. What method was used here? Were they log-10, nL (log), sqrt, transformed?
The authors do not explain why they adjusted for certain covariates and how these were chosen. Did they test for statistical significance between exposure and outcome, did they use directed acyclic graphs, or a priori knowledge from previous research? Reason behind this are especially necessary for BMI and ethnicity.
Again, here it says the models were adjusted by ethnicity, but earlier in the methods, the authors indicate that there was only one ethnicity enrolled.
Results:
Please provide a Table 1 with information on the demographic distribution of the population. What are these specific ethnicities the authors are referring to? What is the mean total CBCL score? How any study participants were missing data?
A mean should not be reported without he accompanying standard deviation.
Children should not be described as “normal.” Please refer to CBCL scores as with “typical development” or describe the children as “typically developing”.
If Mn has been log transformed, then the authors should be providing geometric means, or both geo and arithmetic means, but not arithmetic means alone. Conversely, because we know the data are skewed, the authors could just report the median and IQR without the mean.
Please also provide information on the limit of detection for each measurement. Were all measures above the limit of detection? Considering the minimum is 0, I doubt that.
Figure 3: Please provide the variable label instead of the actual variable name as it is formatted in your dataset.
There should not be both a Table 3 and Figure 3. In R it is easy to create a heat map with embedded correlation coefficients.
Discussion:
By “absorbed” do the authors mean bioaccumulate. If this is what is meant, it is more important to point out how long it takes to metabolite these and why toenails may be a more appropriate biomarker than blood and urine
Author Response
Abstract: Reviewer: In the sentence stating “Children in the second or third clinical group, with an elevated externalized CBCL score, had a much higher mean and median level of Mn as compared to the normal group. The aggression score was significantly associated with Mn concentration and gender,” please provide statistics. If word limits are a barrier, I see several sentences that should probably be combined. Please also provide statistics for the statement “Thought score is not significantly associated with Mn concentration but is significantly lower in boys.” Response: We thank the reviewer for the comment. We now provide these statistics in the abstract.
|
Reviewer: The abstract states that this analysis was adjusted for ethnicity, but the methods section indicates that there was only one ethnic group enrolled in the study.
Response: We thank the reviewer for the comment. There were four Pacific Island ethnicities that were collected and included Samoans, Tongan, Cook Island, and Others. In the methods section we have now added “child Pacific ethnicities includes Samoans, Tongan, Cook Island, and Others (which included Tokelau and Niuean), as determined by the mother at age two years”.
Reviewer: The authors toggle back and forth between presenting their findings in present tense and past tense. Please present all results in past tense.
Response: Thanks for noting this very relevant comment. We have now fixed this so that it reads in past tense.
Reviewer: The abstract also states that the analysis was adjusted for gender, but the methods indicate that it was adjusted for sex. Gender and sex are not the same and should not be used interchangeably. Are the authors referring to sex assigned at birth? If so, sex should be used.
Response: We thank the reviewer for the comment and suggestion. We have now changed it to sex.
Introduction:
Reviewer: Please revise the following sentence to either focus on symptomology or proposed etiology. As it stands, it is too confusing what the sentences is supposed to convey: “interrelation between the brain, emotional, cognitive, genetics and behavioral processes across the lifespan”.
Response: We thank the reviewer for their suggestion. We have now reworded the sentence so that it reads: :IDD effects the developing brain, leading to emotionalcognitive, and behavioural processes across the lifespan [3]”.
Reviewer: Manganese is also an essential element, so it can be hazardous when exposures are too low. This should be conveyed as it is only toxic at high doses.
Response: We thank the reviewer for their comment. We have now added this:
“Manganese at lower levels is essential but at higher levels can be toxic especially in children. Recently, Mn exposure at higher levels has emerged as a developmental neurotoxin”.
Reviewer: The authors suggest that the major source of manganese is occupational exposure, but that’s only in adults. As this study focuses on 9 year old children, that is not a relevant exposure source. The primary source of manganese in children in nutritional intake, particularly infant formula.
Response: We thank the reviewer for their comments. We have now modified this sentence and it now reads: “The major source of Mn exposure in adults is occupational (welders mine ore extraction) [31, 32] and environmental (mine ore, pesticides), however this is not the case in children [26]. Mn exposures in children mostly occurs via inhalation [33] or ingestion via drinking water [34]”.
Reviewer: The introduction currently provides far too much study specific information that could simply be moved to the discussion. Please use the intro to briefly summarize what is known and why this study was conducted.
Response: We thank the reviewer. We have now reduced the introduction and brought it down to discussion as suggested.
Methods:
Reviewer: It is not clear why the authors used MANOVA when the score is ordinal and not continuous.
Response: The original score is continuous; it was analysed as a continuous score in MANOVA with natural log transformation; it is also categorised into ordinal grade (normal/borderline/clinical) and analysed using the generalized linear model.
Reviewer: The utility of network analysis should be described for understanding the reasons for use over other methods of join associations (WQS, g-computation, PCA).
Response: It is used to answer a biological question about the intra-relations of the trace elements and its network, it is not used for identifying groups, casual inference or reducing number of dimensions. The g-computation method could be used in future longitudinal studies.
Reviewer: Additionally, just because the data are coded in an ordinal fashion does not make them ordinal. Have the authors tested the proportional odds assumption to confirm that they data are actually ordinal? If not, they should be modeled as nominally discrete variables.
If the authors are including multiple predictors in the model, that is a “multivariable” model, not “multivariate”.
The authors state the data were transformed, but there are several ways to transform the data. What method was used here? Were they log-10, nL (log), sqrt, transformed?
Response: It is the multiple dependants/response variable used and therefore multivariate analysis is used. The original externalizing score has used the natural log transformation.
Reviewer: The authors do not explain why they adjusted for certain covariates and how these were chosen. Did they test for statistical significance between exposure and outcome, did they use directed acyclic graphs, or a priori knowledge from previous research? Reason behind this are especially necessary for BMI and ethnicity.
Response: The graph model is undirected and only to identify the associations. The identified significant covariates are from previous studies[42,43].
Reviewer: Again, here it says the models were adjusted by ethnicity, but earlier in the methods, the authors indicate that there was only one ethnicity enrolled.
Response: We thank the reviewer for their comments. And we are sorry for the confusion. As mentioned earlier, there were four Pacific Island ethnicities that were collected and included Samoans, Tongan, Cook Island, and Others. In the methods section we have now added “child Pacific ethnicities includes Samoans, Tongan, Cook Island, and Others (which included Tokelau and Niuean), as determined by the mother at age two years”.
Results:
Reviewer: Please provide a Table 1 with information on the demographic distribution of the population. What are these specific ethnicities the authors are referring to? What is the mean total CBCL score? How any study participants were missing data?
Response: We have now added subtitle 3.1 in the result section, which has the first paragraph describing the population distribution, including ethnicities. Ethnicity is not a significant factor associated with externalizing CBCL. The summary statistics of CBCL is also included in this paragraph. There are no missing data in element concentration and five patients had missing CBCL quartiles
Reviewer: A mean should not be reported without he accompanying standard deviation.
Response: This has been revised
Reviewer: Children should not be described as “normal.” Please refer to CBCL scores as with “typical development” or describe the children as “typically developing”.
Response: This is revised in normal CBCL score range- typically developed children
Reviewer: If Mn has been log transformed, then the authors should be providing geometric means, or both geo and arithmetic means, but not arithmetic means alone. Conversely, because we know the data are skewed, the authors could just report the median and IQR without the mean.
Response: Thanks for the suggestions. The reasons of reporting mean is to compare with the other literature. Medians are close to the geometric means. Now we include the geometric mean as well as the arithmetic mean.
Reviewer: Please also provide information on the limit of detection for each measurement. Were all measures above the limit of detection? Considering the minimum is 0, I doubt that.
Response: 0.002 is the minimal value of trace element concentration and it is from Hg. The min value of Mn is 0.03 , the decimal place is now added in table.
Reviewer: Figure 3: Please provide the variable label instead of the actual variable name as it is formatted in your dataset.
Response: Revised.
Reviewer: There should not be both a Table 3 and Figure 3. In R it is easy to create a heat map with embedded correlation coefficients.
Response: Revised.
Discussion:
Reviewer: By “absorbed” do the authors mean bioaccumulate. If this is what is meant, it is more important to point out how long it takes to metabolite these and why toenails may be a more appropriate biomarker than blood and urine
Response: We thank the reviewer for this important comment. We have reworded this to:
“All of these elements at higher levels can be toxic and some like Pb can bioaccumulate in humans”.